# Sensing System Based on FBG for Corrosion Monitoring in Metallic Structures

**DOI:** 10.3390/s22165947

**Published:** 2022-08-09

**Authors:** Israel Sousa, Luis Pereira, Esequiel Mesquita, Vitória L. Souza, Walney S. Araújo, Antônio Cabral, Nélia Alberto, Humberto Varum, Paulo Antunes

**Affiliations:** 1Laboratory of Buildings Durability and Rehabilitation, Campus Russas, Federal University of Ceara, Russas 62900-000, Brazil; 2I3N & Department of Physics, University of Aveiro, Campus Universitário de Santiago, 3810-193 Aveiro, Portugal; 3Department of Metallurgic Engineering, Campus PICI, Federal University of Ceara, Fortaleza 60020-181, Brazil; 4Department of Structural Engineering and Civil Construction, Campus PICI, Federal University of Ceara, Fortaleza 60020-181, Brazil; 5Instituto de Telecomunicações, University of Aveiro, Campus Universitário de Santiago, 3810-193 Aveiro, Portugal; 6CONSTRUCT-LESE, Structural Division, Department of Civil Engineering, Faculty of Engineering, University of Porto, 4200-465 Porto, Portugal

**Keywords:** optical sensors, fiber Bragg gratings, corrosion, metallic structure

## Abstract

As corrosion has slow development, its detection at an early age could be an alternative for reducing costs of structural rehabilitation. Therefore, the employment of structural health monitoring (SHM) systems, sensing configurations collecting data over time allowing for observing changes in the properties of the materials and damage emergence, for monitoring corrosion can be a good strategy to measure the damage and to decide the better moment for intervention. Nonetheless, the current corrosion sensor technology and the high costs of the sensing system implementation are limiting this application in the field. In this work, an optical fiber Bragg grating (FBG)-based sensing system is proposed for monitoring the thickness loss of a 1020 carbon steel metal plate subjected to controlled corrosion. The natural frequency of the plate was collected as a function of the corrosion time over 3744 h. To validate the experimental results, ultrasound measures and electrochemical tests were also carried out under similar conditions. The experimental results show adequate reliability, indicating the suitable functionality of the proposed system for monitoring the thickness loss caused by corrosion in metallic structures, in comparison with traditional methods, as ultrasonic and electrochemical measures.

## 1. Introduction

Corrosion is one of the main damages that act on civil structures, characterized as the deterioration of the material caused by a chemical or electrochemical interaction of the material with the environment [1]. Corrosion can cause serious damage to the structures, resulting in economic, environmental, and worst, loss of life. In addition, corrosion is a phenomenon that can present itself in different ways, usually considering the form of attack and the different mechanisms responsible for its cause [2,3]. From this, corrosion can then be classified according to different factors, such as morphology, causes of origin, the action of mechanical factors, the corrosive environment, and the location of the attack. Morphological classification is of special importance due to the help in identifying the origin of corrosion and in choosing the best alternatives to control and protect structures subject to this phenomenon. This classification includes the most common type of corrosion, the uniform corrosion, which acts on the entire surface area of the metal, promoting a loss of uniform thickness, and pitting corrosion, which is one of the most harmful to structures due to the formation of points that can act, causing perforations and unwanted concentrations of stress in these regions, causing greater fragility to mechanical actions [2,4,5]. Noteworthy as well are intergranular and transgranular corrosion, in which they differ with the corrosion occurring between the grains of the crystal structure in the first and second occurring in the grains of the crystal structure, both causing a loss in the mechanical properties of the corroded material [6,7]. Based on this, the development of new techniques for structures and material characterization of this damage has emerged in recent years.

To analyze the progress of corrosion in metallic structures, some techniques are already commonly used, such as visual examination [8], which consists of the analysis of the formation of corrosion products, allowing the identification of the type of corrosion presented, as well as ultrasonic techniques [9,10], acoustic emission [11,12] and eddy currents [13,14]. All these techniques are based on nondestructive methods, with special interest in the identification of damages, cracks, internal defects, leaks, and others. Noteworthy also is the technique of using corrosion coupons, applied with the objective of determining the corrosion rate [15], as well as the use of electrochemical methods, such as the linear polarization resistance method, also responsible for the possibility of estimating the instantaneous velocity of corrosion in metals [16].

There are some other corrosion monitoring techniques, such as electrical resistance [17,18], where the thickness reduction is measured, allowing for observing the corrosion index. However, this method is limited to the difficulty of detecting localized corrosion. Other techniques for corrosion evaluation are also being proposed, such as the one described in [19], in which a rebar corrosion sensor is embedded in concrete and the monitoring is based on the surface acoustic wave. Nonetheless, these techniques are limited to punctual measurements, have high costs, are time-consuming, have flame risk, and experience electromagnetic interference, since the use of electronic devices in aggressive and flammable environments, such as oil pipelines, may result in an explosion. Thus, the development of alternative monitoring methods for corrosion monitoring has emerged to find more suitable and advantageous sensing solutions. For instance, in [20], radiography techniques are explored for gas pipeline monitoring, and in [21], an ultrasonic-based testing method is proposed. These techniques have, however, limitations, in which the main one consists of carrying out punctual measurements, making it impossible to carry out continuous monitoring [22]. According to [23], even though 858 patents on optical sensors have been registered till 2014 on the main patent offices, representing a high advance on the sensing application, this number is still low in comparison with electronic devices registered in the same period. Ref. [23] highlights that electronic sensing devices have a limitation on their field application due to the electromagnetic field interference in the measurements, while optical sensors can be freely used without electromagnetic interference. This way, the development of new optical sensors can be a great contribution to advance the structural health monitoring (SHM) field.

Compared with electronic sensing systems, in addition to electromagnetic interferences, optical fiber sensors present several advantages, including reduced weight and dimensions, low transmission loss, and high electrical isolation. Additionally, in the case of fiber Bragg gratings (FBGs), these present a high signal/noise ratio, as well as the possibility of sensor multiplexing, allowing the inscription of several fiber gratings in a single optical fiber [24]. Optical sensors are capable of measuring different physical–mechanical, chemical, and environmental parameters of the structures under analysis, with an emphasis on the parameters of the first one, with the capability to measure stress [25], deformation [26], acceleration [27], and temperature [28]. When it comes to chemical properties, the ability to measure relative humidity [29] and pH [30] stands out. As for the environmental parameters, the measurements of wind direction and speed stand out [31], as well as irradiation [32].

In the last years, FBG technology has been applied in the civil and mechanical engineering field [33,34,35,36,37]. Optical sensors have also gained prominence due to the possibility of application in distinct areas. Among the highlighted areas there is SHM, consisting in the use of a technical approach aimed at the structural characterization of engineering constructions [38,39,40,41].

Due to their resistance to aggressive environments, it is possible to use them in oil drilling, as exposed by Qiao et al., who propose FBG sensors to monitor temperature and liquid level in an oil field [42]. There are studies in the literature on the use of optical sensors in the detection of corrosion, such as in reinforced concrete structures [43,44,45], in metallic bridges [46,47], and for the monitoring of corrosion in prestressed structures [48]. Regarding the use of FBGs for the detection of corrosion in the pipelines, Ren et al. [49] proposed a monitoring technique based on the circumferential tension and its relationship with pressure and thickness variations. However, this method is difficult to apply when the pipe has large diameters. The use of FBG sensors to monitor circumferential stresses was also explored by Sun et al. [50]; however, the results obtained did not lead to a quantitative analysis of corrosion, not defining the damage that occurred to the pipeline. The detection of external corrosion on oil pipelines was investigated by Vahdati et al. [51], who proposed an FBG-based sensing system using multiple sensors, each consisting of a semicircular plastic strip (with an embedded FBG to monitor the strip’s stress) and a flat dog-bone-shaped sacrificial metal plate made out of the same pipeline material. As corrosion affects the sacrificial metal plate, the stress on the plastic strip varies, and consequently, the strain is experienced by the FBG sensor. However, this sensor allows only localized severe corrosion detection, as the sacrificial metal plate needs to be affected by corrosion itself in order to detect this damage in the pipeline.

In this work, an FBG-based sensing system is proposed to detect the initial stage of corrosion and to monitor the thickness loss of a 1020 carbon steel metal plate subjected to controlled corrosion using a NaCl solution. The sensing system aims to detect the generalized corrosion since its beginning. The natural frequency variation of the plate is used to monitor the corrosion evolution. To validate the proposed methodology, ultrasound measures and electrochemical tests were also carried out under similar conditions, and the results were compared between them.

## 2. Theory of the Fiber Bragg Gratings

FBGs are periodic modulations of the refractive index along the core of the fiber, normally created through the incidence of an optical pattern of ultraviolet interference. When the FBG is illuminated by a broadband source, it will reflect portion of the spectrum centered on the Bragg wavelength (which satisfies the first-order Bragg condition), while all the other wavelengths will be transmitted. The Bragg condition defines the region of the spectrum centered on the Bragg wavelength that is reflected, and is given by the following equation:(1)$$\lambda_{B}=2\Lambda n_{eff}$$
where *λ_B_* is the reflected Bragg wavelength, *Λ* is the period of the modulation of the refractive index, and *n_eff_* is the effective refractive index of the optical fiber core.

As shown in Equation (1), the reflected wavelength depends primarily on the period of the refractive index modulation and the effective refractive index of the optical fiber core. Thus, when an external parameter, namely, temperature and strain, causes changes in any of them or both, there is a variation of the reflected Bragg wavelength ($\Delta \lambda_{B}$) given by Equation (2). This is the working principle of an FBG as sensor.
(2)$$\Delta \lambda_{B}=2\left(\Lambda \frac{\partial n_{eff}}{\partial T}+n_{eff}\frac{\partial \Lambda }{\partial T}\right)\Delta T+2\left(\Lambda \frac{\partial n_{eff}}{\partial l}+n_{eff}\frac{\partial \Lambda }{\partial l}\right)\Delta l=S_{T}\Delta T+S_{l}\Delta l$$

The first term of the equation represents the influence of temperature on the variation of the Bragg wavelength, while the second part of the equation represents the influence of deformation. $S_{T}$ and $S_{l}$ are the temperature and strain sensitivity coefficients of the FBG sensors.

The principle applied in this work is based on the fact that the advance of corrosion induces a reduction in the mass of the corroded material, and a change in stiffness, causing variations in the natural frequencies of the structure. Equation (3) shows the relationship of these parameters:(3)$$fn=\;\frac{n^{2}\pi }{2}\sqrt{\frac{EI}{mL^{4}}}$$
where *fn* is the natural frequency, *n* is the number of modes, *E* is the plate Young’s modulus, *I* is the moment of inertia of the area, *m* is the mass, and *L* is the plate length.

By applying a vertical impact to the metal plate, the plate oscillates with its natural frequency. This vertical impact causes small deformations in the plate and, consequently, in the fiber that is fixed to it, and is directly related to the variation of the Bragg wavelength, as observed in the second part of Equation (2). On the other hand, the Bragg wavelength also shifts due to temperature variations (first part of Equation (2)), but this measuring method is not directly affected by this issue, and monitoring techniques can be implemented in field applications to nullify cross-sensitivity effects. For instance, the frequency corresponding to temperature variations is usually very different from the natural frequencies of the metallic structures; thus, frequency filters can be used in order to obtain only the data about the natural frequencies. Additionally, by exploring the multiplexing capabilities of these grating devices, FBG-based temperature sensors can also be used to directly calibrate the response of the FBG-based frequency sensors.

## 3. Experimental Procedure

The monitoring of the thickness change caused by corrosion was carried out through three different methods: ultrasound measures, use of FBG based sensors, and electrochemical tests, which will be described in the following subsections.

### 3.1. Ultrasound Measures

A 1020 carbon steel plate, with 3 mm thickness and an area of 18.09 cm^2^, was immersed in 0.1 M NaCl. With a thickness gauge (AK841, Akso, São Leopoldo, Brazil), the thickness variation was verified before the solution application and afterward at times of 1, 3, 6, 24, 168, 336, 504, 744, 840, 1248, and 1344 h.

### 3.2. FBG-Based Sensing

Nine FBGs, with 8 mm physical length, were inscribed into a single-mode GF1 fiber (Thorlabs, Newton, NJ, USA) through the phase mask technique, using a pulsed Q-switched Nd:YAG laser system (LS-2137U Laser, LOTIS TII, Minsk, Belarus) emitting at 266 nm. The FBGs, each spliced to a single-mode pigtail, were distributed and fixed in a 1020 carbon steel, with dimensions of 300 mm × 700 mm and a thickness of 3 mm, according to Figure 1a.

The FBG sensors were fixed with some random strain to the plate using X120 adhesive (HBM, Darmstadt, Germany), where each FBG was embedded in an adhesive layer with 30 mm length and 5 mm width (black areas in the plate). Before gluing the FBG sensors, the protective coating of the steel plate was removed with a paint stripper in nine locations corresponding to the position of each sensor (note that the removed coating sections have the same dimensions as the applied adhesive layers with FBGs embedded to avoid the metal from being exposed to the external environmental conditions).

The corrosion was induced in a region with an area of 40.715 cm^2^ (region “h” in the Figure 1a,b), with a 0.1 M NaCl solution. The protective coating of this region was previously removed with a paint stripper and sanded to guarantee the direct contact of the solution with the metal. A polyvinyl chloride (PVC) pipe section was glued to support the application of the fluid, without leaking along the plate and compromising the performance of the experimental procedure. However, the protective coating of two regions was removed to induce corrosion (see Figure 1b; only the one on the left side was used (h), as shown in the illustrative scheme in Figure 1a).

The optical signal of the sensors was monitored with a customized interrogator composed of an amplified spontaneous emission broadband light source (AS4500 Series from Shanghai B&A Technology, Shanghai, China) and a spectrometer (I-MON 512 USB from Ibsen Photonics, Hadsund, Denmark) with a maximum measurement frequency of 3 kHz, initially without inducing controlled corrosion, and after this procedure, at specific periods, for 3744 h in which the solution acted on the metal.

Bragg wavelength variations were determined by applying the vertical impact to the metal plate. This was determined initially without inducing controlled corrosion and after this procedure, at specific periods, during the corrosion process in which the solution acted on the metal.

### 3.3. Electrochemical Tests

Electrochemical tests were carried out on three 1020 carbon steel sheets samples, with 3 mm of thickness, in which a 2.65 cm diameter PVC pipe was glued, allowing a metallic surface area of 5.51 cm^2^ to be in contact with a 0.1 M NaCl solution. In these tests, an electrochemical cell composed by three electrodes (working electrode (1020 carbon steel), against platinum electrode, and reference electrode Ag(s)/AgCl(s)/Cl-(aq) (saturated KCl)) was used. Electrochemical measurements were performed with the equipment potentiostat/galvanostat model PGSTAT30 (Autolab, Metrohm-Eco Chemie, with data collection made in Nova v.2.1.4 software, Utrecht, The Netherlands). Figure 2 shows the electrochemical cell assembly.

The open-circuit potential (OCP) measurement consisted of monitoring the sample potential as a function of the open-circuit time, without applying potential or current from external sources. A stabilization period of 30 min of the electric potential was considered before the measured reading.

Through electrochemical impedance spectroscopy, it is known that impedance is the resistance that the electrolyte (solution) in contact with the metallic surface imposes on the transfer of electrical current. Electrochemical impedance spectroscopy was then performed to obtain an accurate analysis of the material. Thus, the carbon steel sheet was in contact with the NaCl solution for 24 h, and electrochemical impedance measurements were taken after 30 min, 2 h, 4 h, 6 h, 8 h, and 24 h of immersion. A frequency ranging from 40 kHz to 6 MHz was used, with an amplitude signal of 10 mV.

## 4. Results and Discussion

### 4.1. Ultrasound Measures

The thickness of the metal plate, determined by ultrasound measures, as a function of the corrosion time can be seen in Figure 3.

Through the collected results, the curve corresponding to the variation of the thickness as a function of time was determined, with a nonlinear adjustment coefficient equivalent to 98.99%. The equation that determines this correlation is expressed below, with *T* equivalent to thickness and *t* to time:(4)$$T=337.54+2715.73\times e^{-\frac{t}{384.47}}$$

For the plate used in this experimental procedure, the calculated lost masses (*m_q_*) were determined using the following expression:(5)$$d=\;\frac{m}{V}=\frac{m}{A\times T}$$
where *d* is the density of the 1020 carbon steel, estimated as 7870 kg/m^3^ [52]; *m* is the mass of the region where the 0.1 M NaCl solution was applied; *V* is the volume of that region; and *A* is the area of the section. The volume was estimated as a function of the thickness and area of the section, equivalent to 18.09 cm^2^. The calculated lost mass was then determined by subtracting the mass of the region already reduced due to the corrosive process from the mass of the region before the corrosion process (*m*_0_). The results are presented in Table 1.

The equation that defines the variation of the calculated lost mass (*m_q_*) as function of time (*t*) can be obtained from the relationship between Equations (4) and (5) and given by:(6)$$m_{q}=m_{0}-m=A\times d\times \left(T_{0}-T\right)=38.66-38.66\times e^{-\frac{t}{384.47}}$$
where *T*_0_ is the initial thickness of the plate, calculated from Equation (4), and is equal to 3053.27 μm (*t* = 0 h). Figure 4 compares the variation of the calculated lost mass using Equation (6) (red line) with the obtained values from Table 1 (black squares), showing a good fit with R^2^ = 97.21%.

Carbon steel, in the corrosion process, first gains mass due to the corrosion product formed in the process, and later, there is a reduction in mass after the cleaning process. It may happen, depending on the conditions of the environment, that the advance of corrosion is sufficient for the corrosion product to become adherent and not be removed with acid cleaning, causing the mass to stabilize [2].

It is also noted that the ultrasound measurements can sometimes present some errors due to the procedure involved, such as the variation in the positioning of the transducers and the total contact of the transducers with the surface. This associated error can explain the high variation in thickness observed in the initial moments of the measurement, as shown in Figure 3.

### 4.2. FBG-Based Sensing

Figure 5a shows the response of FBG sensor 1 after the application of a vertical impact, five times during 10 s and 48 h after applying the 0.1 M NaCl solution. Through the fast-Fourier transform (FFT), the natural frequencies of the steel plate were determined, and the obtained results are presented in Figure 5b.

The frequency values found by the nine sensors, before the corrosion process and before gluing the NaCl solution container (PVC pipe section) to the plate, had an average equivalent to 44.0 Hz. After gluing the PVC pipe section to the plate, the frequency decreased by roughly 1 Hz. The experimental results of frequency variation as a function of corrosion action time for the nine sensors can be seen in Figure 6, in which the trend line that determines the behavior of the curve is defined by a logarithmic function.

The coefficient of determination and the curve equation shown in Figure 6 are presented below in Table 2.

The corrosion, caused by the controlled application of the NaCl solution, originated a reduction in the thickness of the metallic plate in its application region and a reduction in the material’s stiffness. This reduction in thickness implies a reduction in mass and, consequently, an increase in the natural frequency, while the reduction in stiffness results in a reduction in the natural frequency. With the increase in the natural frequency as a function of time, as seen in Figure 6, a greater influence of the mass loss on the natural frequency variation is observed, compared with the reduction in stiffness. This effect can be observed mainly in the first instants after the corrosive solution is in contact with the steel plate. A tendency toward the attenuation of the growth of the natural frequency as a function of time in the interval between 144 and 384 h after the NaCl application is also observed. This effect is related to the corrosion speed reduction and the increase in the amount of the corrosion products, such as iron oxides, during the initial 144 h. Between 2328 and 3744 h after the NaCl application, there was a tendency toward a slow increase and stabilization in the natural frequency of the metallic plate. This behavior is associated with the fact that the same solution was used during the entire experiment, and the compositional changes and inhomogeneity of the solution were not suppressed as the corrosion process occurred. Additionally, at this stage of the process, the NaCl solution was already saturated with corrosion products, which decreased drastically the corrosion effect.

Figure 7 presents photographs of the NaCl solution at different times of the corrosion process, with the darkening of the solution being notable. This phenomenon could be attributed to the presence of iron oxide in the collected samples, a product of the corrosion whose concentration increases in the initial moments. It was also observed that 2328 h after the application of the solution, the corrosive liquid seemed clearer, a fact that could be justified by the deposition of a greater number of particles from the corrosion process on the bottom of the container and with a larger size, after 30 min of stabilization time. At other times, these particles were dispersed, making the solution homogeneous. A greater precipitation of the solution was also observed as a result of its evaporation occurring during the corrosive process.

According to Figure 6, the sensor responses present a nonlinear curve fit suitable for the predetermined needs. The instability of some sensors (particularly the ones with a lower coefficient of determination) and the offset between measurement values from different FBG sensors during the experiment can be explained by the fact that the FBGs were prestrained manually (with random strain) in the installation process, using duct tape to fix the fiber during the cure of the X120 adhesive, and consequently, relaxation of the fiber can occur. From the combination of results, it was possible to relate the plate calculated lost mass with the natural frequency variation. Thereunto, a *q* factor was estimated, equivalent to the ratio between the calculated lost mass and the area of the region subject to corrosion by the NaCl solution (from Section 4.1), equivalent to 18.09 cm^2^, for each of the determined times. These values can be seen in Table 3.

The relationship between the *q* ratio values from Table 3 and corrosion time can be seen in Figure 8 (black squares). Since the *q* values for 3 and 6 h after the application of the solution were too small, they were adopted as null, whereas the value corresponded to 24 h after the application of the NaCl was adopted in the module, as 24 h after the application of the solution, corrosion became evident. The curve that represents the variation of the *q* factor in a function of time is obtained from Equation (6) and is given by the following equation:(7)$$q=2.14-2.14\times e^{-\frac{t}{384.47}}\;$$

Figure 8 also depicts the variation of the *q* factor given in Equation (7) (red line), and the nonlinear adjustment coefficient found between the curve and the values from Table 3 was equivalent to 98.23%.

To estimate the calculated lost mass values from the steel plate monitored by the FBG sensors, the *q* (Equation (7)) was multiplied by the area of the region subjected to corrosion on that plate, equivalent to 40.715 cm^2^. Therefore, the relationship between the calculated lost mass and the corrosion time for this plate can be described by:(8)$$m_{q}=\left(2.14-2.14\times e^{-\frac{t}{384.47}}\;\right)\times 40.715\;$$

The relationship between the measured natural frequency and the calculated lost mass (using Equation (8)) for each of the nine FBG sensors is presented in Figure 9.

A logarithmic function was defined for each one of the nine sensors relating the frequency and the calculated lost mass, and this correlation can be seen in Table 4. The linear adjustment for the first 120 h in which the corrosion process occurred is also presented in Table 4.

Similar to the results obtained in Figure 6 and Table 2, Sensor 1 presents the highest coefficient of determination in both circumstances, with a value of 89.50% during the entire experiment (logarithmic function) and a value of 90.27% during the first 120 h (linear function). The obtained results from the FBG sensors, for this steel plate, show that during the initial 120 h, the natural frequency increased by an average of 0.05 Hz for each lost gram.

Therefore, all sensors, even with different positioning and distances from the corrosion point, proved to be sufficiently capable of measuring the variation of the natural frequency.

### 4.3. Electrochemical Tests

Figure 10 contains the measurements obtained with the OCP for the three samples, showing that the reduction of the potential as a function of time occurs for all samples, indicating the presence of a corrosive process.

Figure 11 shows the Nyquist diagrams used to represent the electrochemical impedance results for the three samples. 

The electrochemical impedance reaction may be suitable for an equivalent electrical circuit. Knowing that the horizontal axis of the Nyquist diagram represents the real impedance of the equivalent circuit and the vertical axis of the diagram corresponds to the imaginary (complex) impedance, it is possible, by the behavior of the semicircle, to infer whether the material is more or less resistant to corrosion, analyzing mainly the horizontal axis.

Thus, for sample 1, there was greater resistance with 24 h of immersion, followed by 8 h, 2 h, 6 h, 4 h, and 30 min. Concerning sample 2, the greatest resistance was obtained at 8 h, followed by 4 h, 24 h, 2 h, 6 h, and 30 min. For sample 3, the highest value found was also at 8 h, followed by 6 h, 24 h, 4 h, 30 min, and 2 h.

Although it is expected that the material will present less resistance to corrosion over time, this may not necessarily occur, as, during the chemical reaction of the corrosive process, an oxide layer (in this case, iron oxide) is formed, which can work as a barrier to the advancement of the corrosive process. Thus, there is an increase in resistance at that moment, which may explain the results obtained from the Nyquist diagrams shown in Figure 11. Therefore, the proposed system makes it possible, through the natural frequency, to monitor precisely, in the order of micrometers, the evaluation of the corrosion process in metallic structures.

Based on the obtained results, the proposed optical sensing system has great potential for in situ applications, especially in possibly inflammable and explosive environments. The environmental vibrations or external impacts may produce the excitation necessary to identify the natural frequencies. Considering that corrosion (and other types of structural damage) causes small changes in the natural frequency of a metallic structure, the natural frequency variation can be interpreted as the appearance and progress of damage over time. The FBGs are a viable and feasible solution in monitoring the structural frequencies, and this detection method can be combined very effectively with other punctual-based corrosion monitoring techniques, since by monitoring the natural frequencies, it is possible to obtain a general analysis of the presence of damage in the structure, which can later be examined with techniques of greater precision and mapping capability in order to assess the type of damage and its extent.

## 5. Conclusions

In this work, the development of an optical FBG-based sensing system is proposed for monitoring the integrity of metallic structural elements in a continuous way. The methodological approach used consisted in the analysis of the natural frequency variation with time of elements subjected to controlled corrosion under NaCl solution.

Thus, the responses of the nine sensors were adequate for the proposed objective, since in all of them, it was possible to observe a pattern of behavior of the curve defined by the variation of the natural frequency as a function of time, namely, for sensors 1 and 8, in which coefficients of determination were observed to be equivalent to 90.50% and 82.70%, respectively, to the most suitable logarithmic function that describes the frequency variation. Sensors 3, 7, and 9 presented coefficients of determination equivalent to 74.70%, 74.02%, and 74.45%. Sensors 2, 4, 5, and 6 showed a coefficient of determination below 70%, with 62.87%, 66.42%, 68.20%, and 69.69%, respectively, evidencing that even with different positioning and distances, all sensors were able to measure the variation of the natural frequency of the metal plate.

The correlation between natural frequency and calculated lost mass showed a logarithmic behavior and presented coefficients of determination equivalent to 89.50%, 79.69%, 81.53%, and 86.83% for sensors 1, 3, 7, and 8, respectively. For sensors 2, 4, 5, 6, and 9, coefficients of determination were observed to be equivalent to 68.40%, 73.91%, 74.79%, 67.75%, and 76.70%, respectively. Additionally, during the initial 120 h, it was estimated that the natural frequency of the steel plate increased by approximately 0.05 Hz/g.

To validate the data, electrochemical and ultrasound tests were also carried out. Through an ultrasound test, it was possible to estimate the lost mass in the corrosion process. Through the OCP in three samples, it was possible to observe the occurrence of corrosion; then, by performing the electrochemical impedance spectroscopy test, it was possible to infer the variations in the corrosion resistance of the material used. Initially, for sample 1, the highest resistance found corresponded to 24 h after the solution application, while samples 2 and 3 presented a greater resistance to 8 h after contact with NaCl.

Therefore, the results obtained confirmed the reliability of the developed sensing system, enabling its application to assist in decision making in industrial environments and other sectors of the infrastructure, as well as ensuring the guarantee of security due to its capability of application in hostile environments.

## Figures and Tables

**Figure 1 sensors-22-05947-f001:**
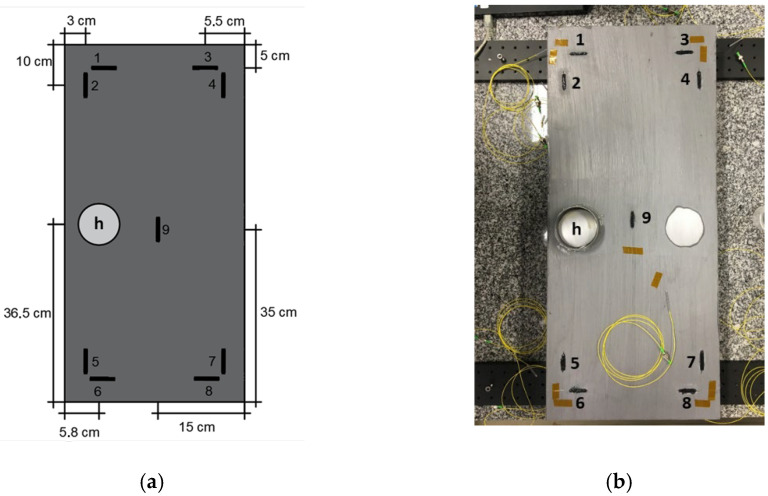
(**a**) FBG sensors’ position and identification and location of the corrosion site on the metal plate; (**b**) photograph of the experimental apparatus.

**Figure 2 sensors-22-05947-f002:**
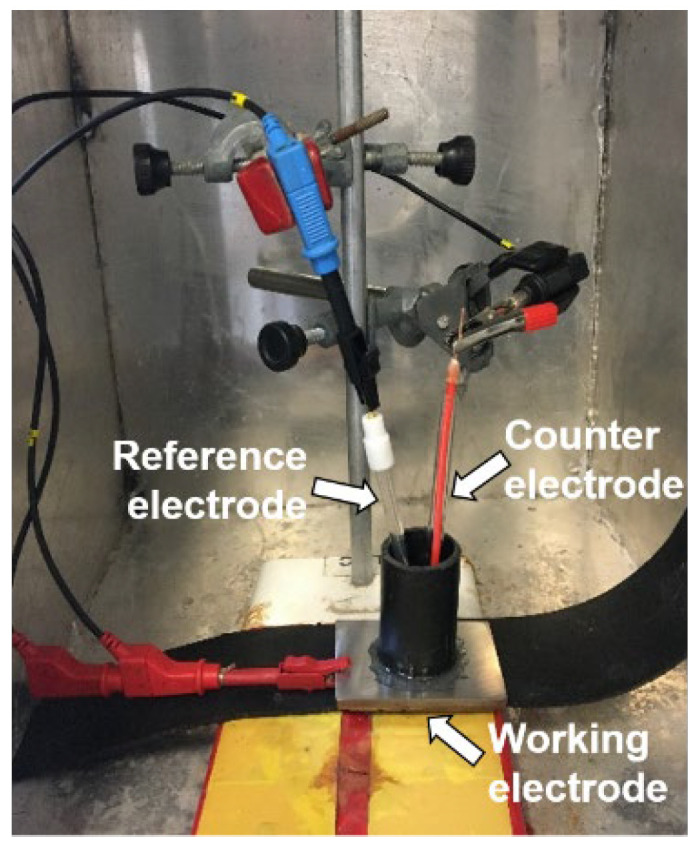
Experimental setup of the electrochemical cell.

**Figure 3 sensors-22-05947-f003:**
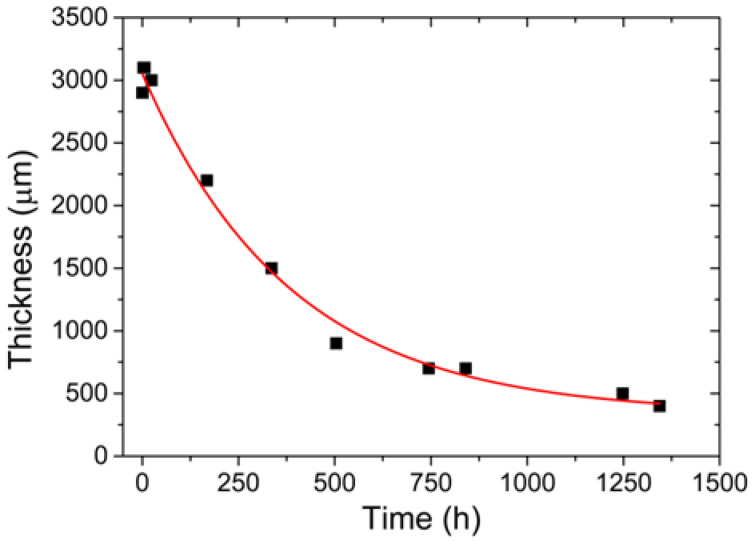
Thickness variation of the metal plate as a function of the corrosion time.

**Figure 4 sensors-22-05947-f004:**
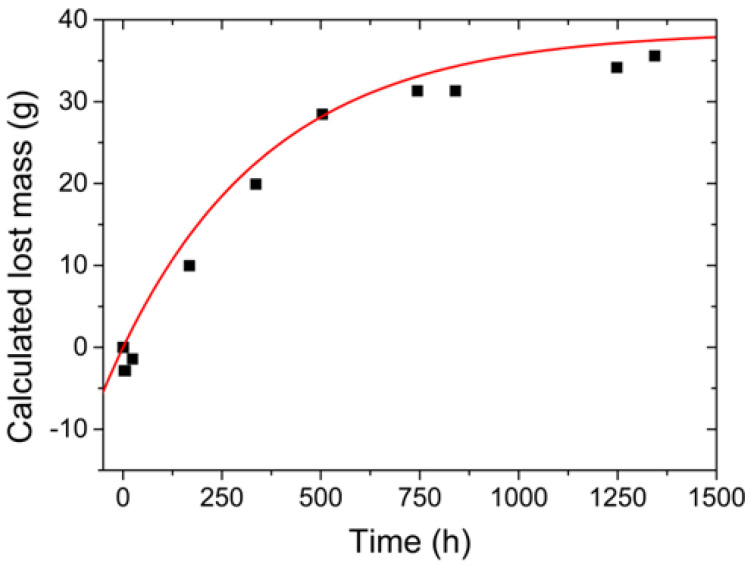
Variation of the calculated lost mass with time using Equation (6) (red line) and the obtained values from Table 1 (black squares).

**Figure 5 sensors-22-05947-f005:**
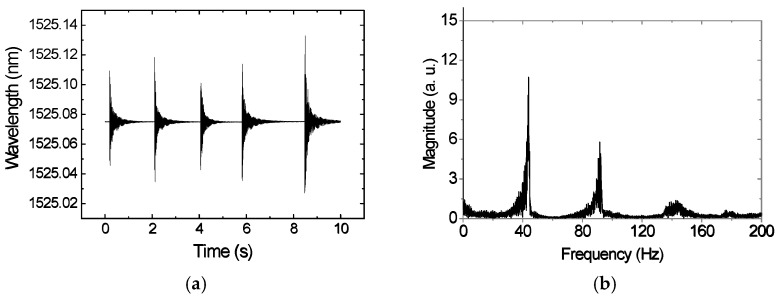
(**a**) Sensor 1 response to the application of five vertical impacts, 48 h after applying the 0.1 M NaCl solution; (**b**) corresponding frequency spectrum after FFT processing.

**Figure 6 sensors-22-05947-f006:**
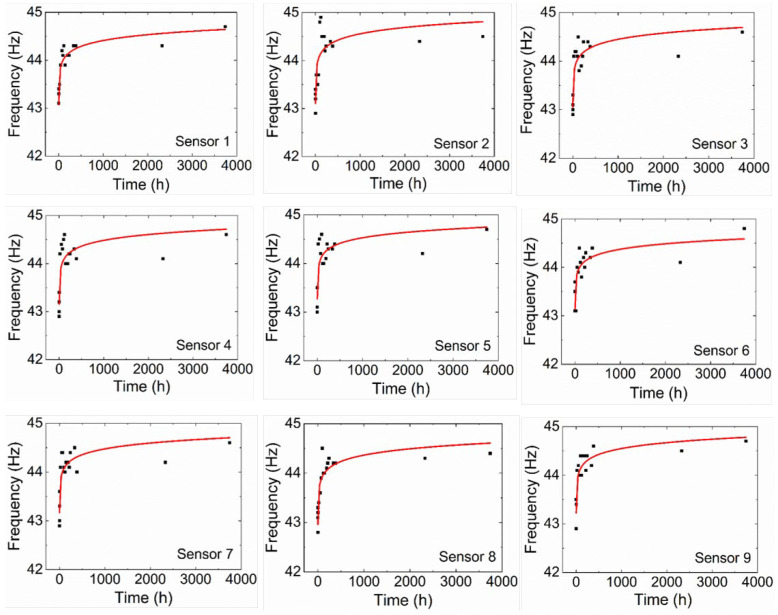
Natural frequency variation as a function of time of the corrosion process on the metal plate for the nine FBG-based sensors.

**Figure 7 sensors-22-05947-f007:**
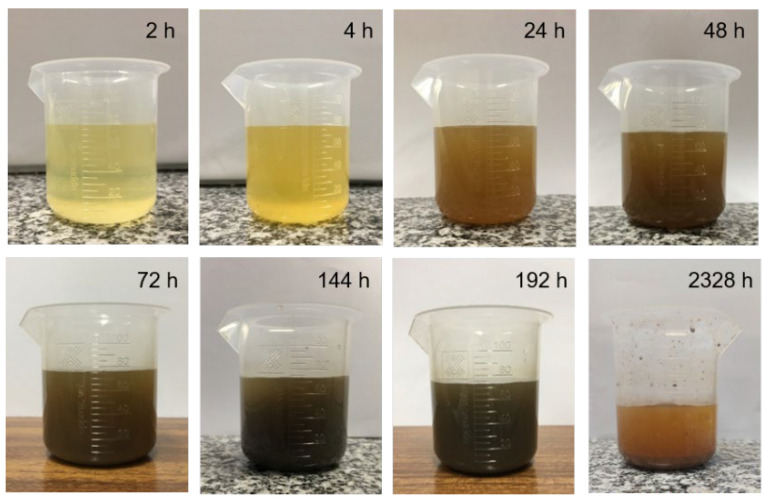
NaCl solution containing the steel plate corrosion products at different times.

**Figure 8 sensors-22-05947-f008:**
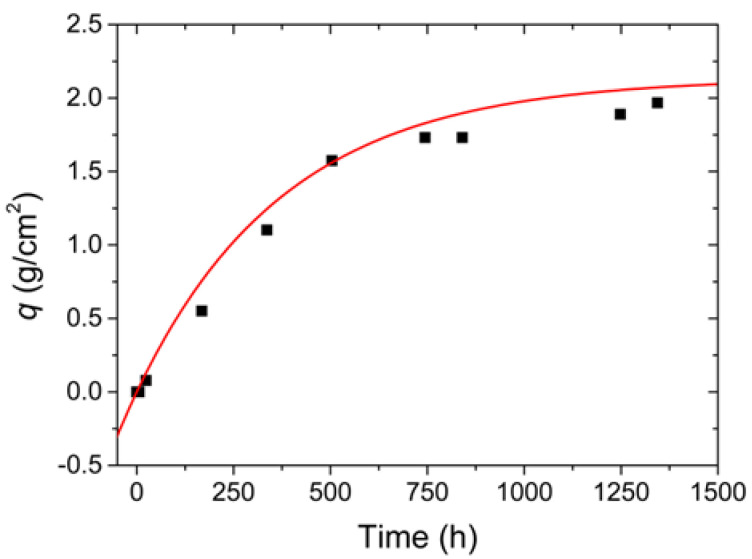
Variation of the calculated lost mass with the corrosion action time using Equation (7) (red line) and the obtained values from Table 3 (black squares).

**Figure 9 sensors-22-05947-f009:**
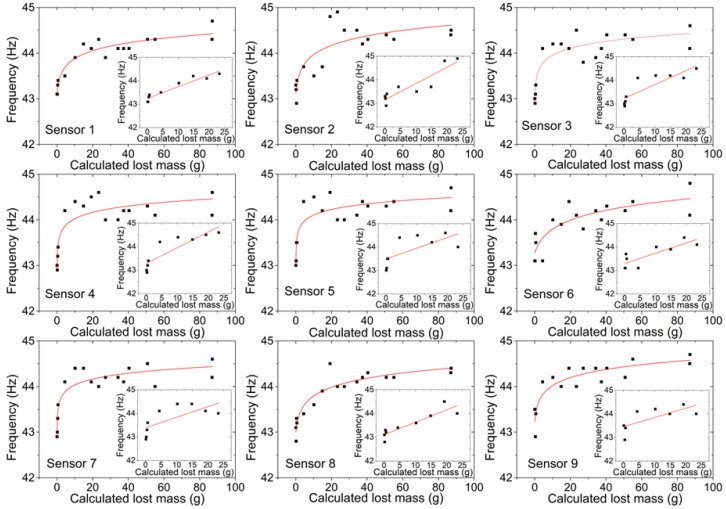
Relation between the calculated lost mass and the natural frequency of the plate obtained by each FBG sensor during the entire experiment. Inset images: Natural frequency variation as a function of the calculated lost mass for each FBG sensor during the initial 120 h.

**Figure 10 sensors-22-05947-f010:**
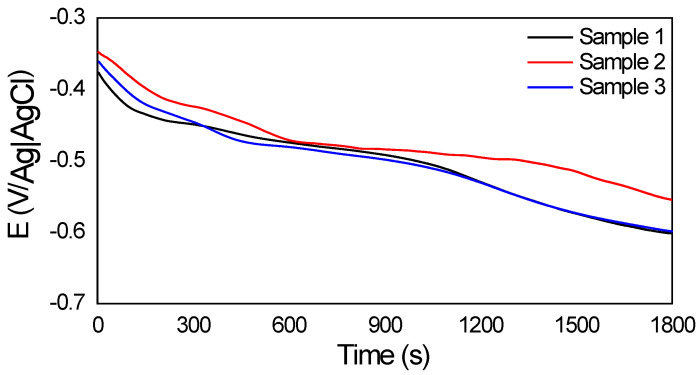
OCP curves for samples 1, 2, and 3 of 1020 carbon steel in NaCl solution.

**Figure 11 sensors-22-05947-f011:**
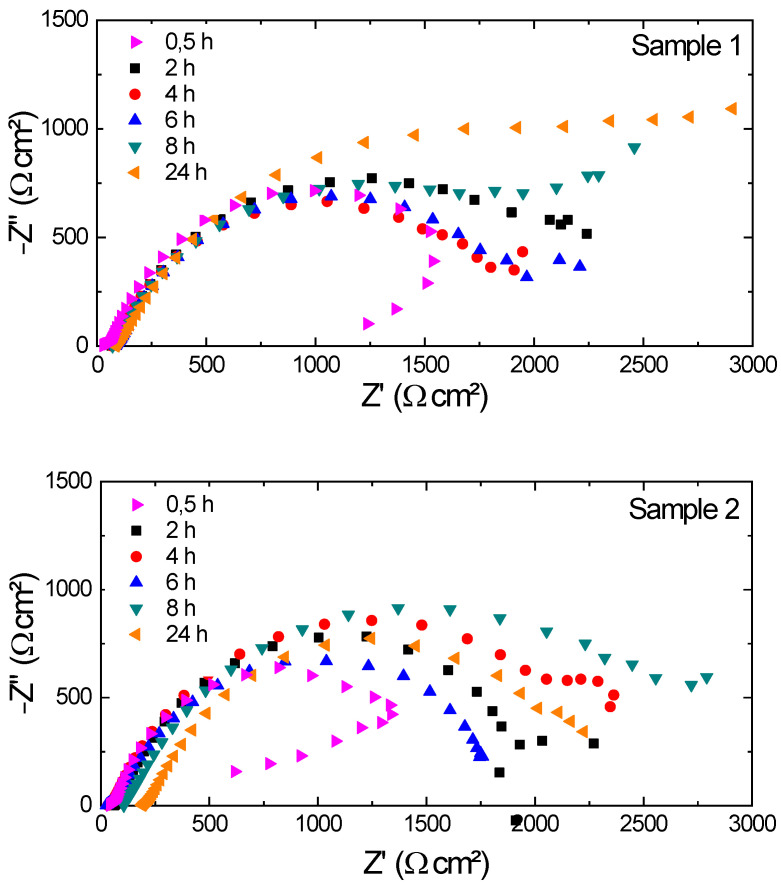
Nyquist diagrams for the three samples of 1020 carbon steel immersed in NaCl.

**Table 1 sensors-22-05947-t001:** Determination of calculated lost mass through ultrasound measurements.

Time (h)	Thickness (μm)	Mass (g)	Calculated Lost Mass (g)
0	2900	41.29	0.00
1	2900	41.29	0.00
3	3100	44.13	−2.85
6	3100	44.13	−2.85
24	3000	42.71	−1.42
168	2200	31.32	9.97
336	1500	21.36	19.93
504	900	12.81	28.47
744	700	9.96	31.32
840	700	9.96	31.32
1248	500	7.12	34.17
1344	400	5.69	35.59

**Table 2 sensors-22-05947-t002:** Results observed in the correlation between frequency and time.

Sensor	R^2^ (%)	Frequency (Hz)
1	90.50	0.18ln(t)+43.19
2	62.87	0.19ln(t)+43.24
3	74.70	0.18ln(t)+43.18
4	66.42	0.17ln(t)+43.28
5	68.20	0.16ln(t)+43.38
6	69.69	0.16ln(t)+43.25
7	74.02	0.17ln(t)+43.29
8	82.70	0.18ln(t)+43.09
9	74.45	0.17ln(t)+43.34

**Table 3 sensors-22-05947-t003:** Determination of the *q* factor, the ratio between the calculated lost mass and the area of the application region of the corrosive solution.

Time (h)	Thickness (μm)	Mass (g)	Calculated Lost Mass (g)	*q* (g/cm^2^)
0	2900	41.29	0.00	0.00
1	2900	41.29	0.00	0.00
3	3100	44.13	−2.85	−0.16
6	3100	44.13	−2.85	−0.16
24	3000	42.71	−1.42	−0.08
168	2200	31.32	9.97	0.55
336	1500	21.36	19.93	1.10
504	900	12.81	28.47	1.57
744	700	9.96	31.32	1.73
840	700	9.96	31.32	1.73
1248	500	7.12	34.17	1.89
1344	400	5.69	35.59	1.97

**Table 4 sensors-22-05947-t004:** Results observed in the correlation between frequency and calculated lost mass.

	During 3744 h		During Initial 120 h	
Sensor	R^2^ (%)	Frequency (Hz)	R^2^ (%)	Slope (Hz/g)
1	89.50	$43.29+0.25\times ln\;\left(m_{q}+0.39\right)$	90.27	0.05
2	68.40	$43.25+0.31\times ln\left(m_{q}+0.56\right)$	79.76	0.07
3	79.69	$43.45+0.22\times ln\left(m_{q}-0.04\right)$	70.74	0.06
4	73.91	$43.57+0.20\times ln\left(m_{q}-0.08\right)$	73.34	0.07
5	74.79	$43.70+0.18\times ln\left(m_{q}-0.10\right)$	39.51	0.05
6	67.75	$42.92+0.34\times ln\left(m_{q}+2.67\right)$	63.80	0.04
7	81.53	$\;43.61+0.19\times ln\left(m_{q}-0.10\right)\;$	42.13	0.04
8	86.83	$43.11+0.29\times ln\left(m_{q}+0.57\right)$	79.39	0.05
9	76.70	$43.41+0.26\times ln\left(m_{q}+0.41\right)$	43.13	0.04

## Data Availability

Not applicable.

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
