# Peer review of "Sensing System Based on FBG for Corrosion Monitoring in Metallic Structures"

_sensors, 2022, doi:10.3390/s22165947_

Round 1
Reviewer 1 Report
I support its publication in this journal when the following issue was solved:
(1) Eq. (4) can be expressed as T= T0exp(-t/t) ? t=1/(0.00193) can be understood as a time parameter about corrosion speed?
(2) Eq. (4) gives the thickness as a function of time, by which, one can derive a loss mass formula, which are seemly different from Eq. (6)?
Reviewer 2 Report
In this manuscript, the authors proposed an optical fiber Bragg grating (FBG) based sensing system for monitoring the thickness loss of metallic structures subjected to corrosion. It presented some experimental results, but the sensing theory of the proposed method is vague. I don't think it has reached the level of publication in Sensors.
1 The main deficiency of the paper is lack of corrosion monitoring theory of the FBG based sensing system. The authors present general knowledge about FBG and basic equations of strain and temperature sensing. But they didn’t provide any meaningful connection between this basic theory and corrosion parameters.
2 As we know there are different forms of corrosions, such as uniform corrosion, local corrosion, pitting corrosion. The author should clarify the scope of application of the proposed method. In addition, in view of the experimental set-up, corrosion occurs in a circular region, and this region is placed in the middle of two sensing area. It is foreseeable when corrosion expanded, the natural frequency of FBG varies. So the question is how about the response if the corrosion doesn’t occur in the middle of the sensing area. The author should discuss about this.
3 Based on the above comments, another confusion is how to implement this method for in situ application. According to the experiment, it seems that the corrosion location should firstly be identified and then installed the FBG sensor around the corrosion region with equal space. I think it is difficult to implement for real application. The author should also clarify this.
Round 2
Reviewer 2 Report
The authors addressed most of my concerns from last reviews. There is still one more issue that I suggest them to modify. From the basic sensing theory of FBG, as the author stated in Eq. (2), the wavelength will be influenced by temperature variation. But throughout the paper, the author didn’t present anything about temperature influence. I know that they just implemented the laboratory test so the temperature can be ignored. But for field application, this may greatly influence the corrosion detection accuracy. I suggest the author to discuss about this issue. Once they are done, this paper is recommended for publication.
Author Response
Reviewer: 2
Recommendations for the Authors:
The authors addressed most of my concerns from last reviews. There is still one more issue that I suggest them to modify. From the basic sensing theory of FBG, as the author stated in Eq. (2), the wavelength will be influenced by temperature variation. But throughout the paper, the author didn’t present anything about temperature influence. I know that they just implemented the laboratory test so the temperature can be ignored. But for field application, this may greatly influence the corrosion detection accuracy. I suggest the author to discuss about this issue. Once they are done, this paper is recommended for publication.
Answer and changes included: Thank you for your comments. We agree with the Reviewer and the necessary changes were done in the manuscript to address this issue. The following commentary was added to the manuscript:
On the other hand, the Bragg wavelength also shifts due to temperature variations (first part of equation (2)), but this measuring method is not directly affected by this issue and monitoring techniques can be implemented in field applications to nullify cross-sensitivity effects. For instance, the frequency corresponding to temperature variations is usually very different from the natural frequencies of the metallic structures, thus frequency filters can be used in order to obtain only the data about the natural frequencies. Also, by exploring the multiplexing capabilities of these grating devices, FBG-based temperature sensors can also be used to directly calibrate the response of the FBG-based frequency sensors.
